# Complexity control by gradient descent in deep networks

Tomaso Poggio[1 ✉], Qianli Liao[1] & Andrzej Banburski[1]

Overparametrized deep networks predict well, despite the lack of an explicit complexity control during training, such as an explicit regularization term. For exponential-type loss functions, we solve this puzzle by showing an effective regularization effect of gradient descent in terms of the normalized weights that are relevant for classification.

[1] Center for Brains, Minds, and Machines, MIT, Cambridge, Massachusetts, USA. ✉email: tp@ai.mit.edu

Once upon a time, models needed more data than parameters to provide a meaningful fitting. Deep networks seem to avoid this basic constraint. In fact, more weights than data is the standard situation for deep-learning networks that typically fit the data and still show good predictive performance on new data[1]. Of course, it has been known for some time that the key to good predictive performance is controlling the complexity of the network and not simply the raw number of its parameters. The complexity of the network depends on appropriate measures of complexity of the space of functions realized by the network such as VC dimension, covering numbers and Rademacher numbers. Complexity can be controlled during optimization by imposing a constraint, often under the form of a regularization penalty, on the norm of the weights, as all the notions of complexity listed above depend on it. The problem is that there is no obvious control of complexity in the training of deep networks! This has given an aura of magic to deep learning and has contributed to the belief that classical learning theory does not hold for deep networks.

In the case of regression for shallow linear networks such as kernel machines, it is well known from work on inverse problems and in machine learning (see refs. [2,3]) that iterative gradient descent (GD) has a vanishing regularizing effect with the iteration number $t$ (for fixed step size) being equivalent to $\frac{1}{\lambda}$ (where $\lambda$ is the corresponding regularization parameter): thus $t \to \infty$ corresponds to $\lambda \to 0$. The simplest example is least-square regression on a linear network, where vanishing regularization unifies both the overdetermined and the underdetermined cases as follows

$$\min_{w \in R^d} \frac{1}{n} ||Y - Xw||^2 + \lambda ||w||^2 \qquad (1)$$

yielding $w_\lambda = (X^T X + \lambda n I)^{-1} X^T Y$. It is noteworthy that $\lim_{\lambda \to 0} w_\lambda = w^\dagger$ is the pseudoinverse. In this case, iterative GD minimizing $\frac{1}{n} ||Y - Xw||^2$ performs an implicit regularization equivalent to taking $\lambda \to 0$ in $w_\lambda$ above.

The question is whether an effective regularization may be similarly induced in nonlinear deep networks by GD and how. This paper addresses the question of implicit regularization in the specific case of training wrt exponential-type loss functions—such as cross-entropy or exponential loss. It is worth noting that cross-entropy is the loss function used in training deep networks for classification problems, and that most of the practical applications and successes of deep networks, at least so far, involve classification.

This paper answers in the affirmative: there is a hidden regularization in typical training of deep networks. The basic observation is that the weights computed during minimization of the exponential loss do not matter by themselves and in fact they diverge during GD. As shown in the next section, in the case of classification—both binary and multiclass—only the normalized weights matter: thus, complexity control is implicit in defining the normalized weights $V_k$ as the variables of interest. What is not completely obvious is that commonly used GD on the unnormalized weights induces a dynamics of the normalized weights, which converges to a stationary point. We show that at any finite time of the dynamics, the stationary points of the flow of $V_k$ satisfy the necessary and sufficient conditions for a minimum. This mechanism underlies regularization in deep networks for exponential losses and, as we discuss later, is likely to be the starting point to explain their prediction performance.

## Results

**Loss functions**. We consider for simplicity of exposition the case of binary classification. We call "loss" the measure of performance of the network $f$ on a training set $S = \{(x_1, y_1), \cdots, (x_N, y_N)\}$. The most common loss optimized during training for binary classification is the logistic loss $L(f) = \frac{1}{N} \sum_{n=1}^{N} \ln(1 + e^{-y_n f(x_n)})$. We focus on the closely related, simpler exponential loss $L(f(w)) = \sum_{n=1}^{N} e^{-f(x_n) y_n}$. We call classification "error" $\frac{1}{N} \sum_{n=1}^{N} H(-y_n(f(x_n)))$, where $y$ is binary ($y \in \{-1, +1\}$) and $H$ is the Heaviside function with $H(-yf(x)) = 1$ if $-yf > 0$, which correspond to the wrong classification. We say that $f$ separates the data if $y_n f(x_n) > 0, \forall n$. We will typically assume that GD at some $t > T_0$ will reach an $f$ that separates the data (which is usually the case for overparametrized deep networks). There is a close relation between the exponential or logistic loss and the classification error: both the exponential and the logistic losses are upper bounds for the classification error. Minimizing the exponential or logistic loss implies minimizing the classification error. Minimization of the loss can be performed by GD techniques. In today's praxis, stochastic GD (SGD) is used to train deep networks. We focus here on GD for simplicity. Our main results should also hold for SGD.

**Deep networks**. We define a deep network with $K$ layers and coordinate-wise scalar activation functions $\sigma(z)$: $\mathbf{R} \to \mathbf{R}$ as the set of functions $f(W; x) = \sigma(W^K \sigma(W^{K-1} \cdots \sigma(W^1 x)))$, where the input is $x \in \mathbf{R}^d$, the weights are given by the matrices $W^k$, one per layer, with matching dimensions. The symbol $W$ is used to denote the set of $W^k$ matrices $k = 1, \cdots, K$. For simplicity, we consider here the case of binary classification in which $f$ takes scalar values, implying that the last layer matrix $W^K$ is $W^K \in \mathbf{R}^{1,K_l}$. As mentioned, the labels are $y_n \in \{-1, 1\}$. The weights of hidden layer $l$ are collected in a matrix of size $h_l \times h_l$ $_{-1}$. There are no biases apart from the input layer where the bias is instantiated by one of the input dimensions being a constant. The activation function is the Rectified Linear Unit (ReLU) activation. For ReLU activations, the following important positive one-homogeneity property holds $\sigma(z) = \frac{\partial \sigma(z)}{\partial z} z$. For the network, homogeneity implies $f_W(x) = \prod_{k=1}^{K} \rho_k f(V_1, \cdots V_K; x_n)$, where $W_k = \rho_k V_k$.

The network is a function $f(x) = f(W_1, \cdots, W_K; x)$ where $x$ is the input and the weight matrices $W_k$ are the parameters. We define the normalized network $\tilde{f}$ as $f_V = \rho \tilde{f}$, with $||V_k|| = 1, \rho = \prod_{k=1}^{K} \rho_k$; $\tilde{\mathbb{F}}$ is the associated class of "normalized neural networks" $\tilde{f}(x)$. It is noteworthy that the definitions of $\rho_k$, $V_k$, and $\tilde{f}$ all depend on the choice of the norm used in normalization. It is also worth noting that because of homogeneity of the ReLU network $f(x) = \rho \tilde{f}(x)$, the signs of $f$ and $\tilde{f}$ are the same.

For simplicity of notation we consider for each weight matrix $V_k$ the corresponding "vectorized" representation in terms of vectors $W_k^{i,j} = W_k$ for each $k$ layer.

We use the following definitions and properties (for a vector $w$ and the 2-norm) neglecting indeces:

- Define $\frac{w}{||w||_2} = v$; thus $w = ||w||_2 v$ with $||v||_2 = 1$.
- The following relations are easy to check:
  1. $\frac{\partial ||w||_2}{\partial w} = v$
  2. $S = I - vv^T = I - \frac{ww^T}{||w||_2^2}$.
  3. $\frac{\partial v}{\partial w} = \frac{S}{||w||_2}$.

**Training by unconstrained gradient descent**. Consider the typical training of deep networks. The exponential loss (more in general an exponential-type loss such as the cross-entropy) is

minimized by GD. The gradient dynamics is given by

$$\dot{W}_k^{i,j} = -\frac{\partial L}{\partial W_k^{i,j}} = \sum_{n=1}^{N} y_n \left[ \frac{\partial f(x_n)}{\partial W_k^{i,j}} \right] e^{-y_n f(x_n)}. \quad (2)$$

Clearly there is no explicit regularization or norm control in the typical GD dynamics of Eq. (2). Assuming that for $T > T_0$ GD achieves separability, the empirical loss goes to zero and the norms of the weights $\rho_k = ||W_k||_2$ grow to infinity $\forall k$. For classification, however, only the normalized network outputs matter because of the softmax operation.

**Training by constrained gradient descent.** Let us contrast the typical GD above with a classical approach that uses complexity control. In this case the goal is to minimize $L(f_W) = \sum_{n=1}^{N} e^{-f_W(x_n)y_n} = \sum_{n=1}^{N} e^{-\rho f_V(x_n)y_n}$, with $\rho = \prod \rho_k$, subject to $||V_k||_p^p = 1 \forall k$, that is under a unit norm constraint for the weight matrix at each layer (if $p = 2$ then $\sum_{i,j} (V_k)_{i,j}^2 = 1$ is the Frobenius norm). It is noteworthy that the relevant function is not $f(x)$ and the associated $W_k$, but rather $f_V(x)$, and the associated $V_k$ as the normalized network $\tilde{f}(x)$ is what matters for classification, and other key properties such as the margin depend on it.

In terms of $\rho$, $V_k$, the unconstrained gradient of $L$ gives

$$\dot{\rho}_k = -V_k^T \frac{\partial L}{\partial W_k} \quad \dot{V}_k = -\rho_k \frac{\partial L}{\partial W_k} \quad (3)$$

as $\dot{\rho}_k = -\frac{\partial L}{\partial \rho_k} = -\frac{\partial W_k}{\partial \rho_k} \frac{\partial L}{\partial W_k}$ and $\dot{V}_k = -\frac{\partial L}{\partial V_k} = -\frac{\partial W_k}{\partial V_k} \frac{\partial L}{\partial W_k}$.

There are several ways to enforce the unit norm constraint on the dynamics of $V_k$. The most obvious one consists of Lagrange multipliers. We use an equivalent technique, which is also equivalent to natural gradient, called tangent gradient transformation[4] of a gradient increment $g(t)$ into $Sg(t)$. For a unit $L_2$ norm constraint, the projector $S = I - \frac{uu^T}{||u||_2^2}$ enforces the unit norm constraint. According to theorem 1 in ref. [4], the dynamical system $\dot{u} = Sg$ with $||u(0)||_2 = 1$ describes the flow of a vector $u$ that satisfies $||u(t)||_2 = 1$ for all $t \geq 0$. Applying the tangent gradient transformation to $\frac{\partial L}{\partial V_k}$ yields

$$\dot{\rho}_k = -V_k^T \frac{\partial L}{\partial W_k} \quad \dot{V}_k = -S\rho_k \frac{\partial L}{\partial W_k} \quad (4)$$

with $S_k = I - \frac{V_k V_k^T}{||V_k||_2^2}$. It is relatively easy to check (see ref. [5]) that the dynamics of Eq. (4) is the same as of the weight normalization algorithm, originally[6] defined for each layer in terms of $w = g \frac{v}{||v||}$, as

$$\dot{g} = \frac{v}{||v||} \frac{\partial L}{\partial w}, \quad \dot{v} = \frac{g}{||v||} S \frac{\partial L}{\partial w} \quad (5)$$

with $S = I - \frac{vv^T}{||v||^2}$. The reason Eqs. (4) and (5) are equivalent is because $||v||^2$ in Eq. (5) can be shown to be constant in time[5]. Weight normalization and the closely related batch normalization technique are in common use for training deep networks. Empirically, they behave similar to unconstrained GD with some advantages especially for very deep networks. Our derivation, however, seems to suggest that they could be different, as weight normalization enforces an explicit, although so far unrecognized, unit norm constraint (on the $V_k$ dynamics), which unconstrained GD (Eq. (2)) seems not to enforce.

**Implicit complexity control.** The first step in solving the puzzle is to reparametrize Eq. (2) in terms of $V_k$, $\rho_k$ with $W_k^{i,j} = \rho_k V_k^{i,j}$, and $||V_k||_2 = 1$ at convergence. Following the chain rule for the time derivatives, the dynamics for $W_k$, Eq. (2), is identical to the

following dynamics for $\rho_k = ||W_k||$ and $V_k$:

$$\dot{\rho}_k = V_k \dot{W}_k \quad \dot{V}_k = \frac{S_k}{\rho_k} \dot{W}_k \quad (6)$$

where $S_k = I - V_k V_k^T$ emerges this time from the change of variables, as $\frac{\partial V_k}{\partial W_k} = \frac{S}{\rho_k}$. Inspection of the equation for $\dot{V}_k$ shows that there is a unit constraint on the $L_2$ norm of $V_k$, because of the presence of $S$: in fact, a tangent gradient transformation on $\dot{V}_k$ would not change the dynamics, as $S$ is a projector and $S^2 = S$. Consistently with this conclusion, unconstrained GD has the same critical points for $V_k$ as weight normalization but a somewhat different dynamics: in the one-layer case, weight normalization is

$$\dot{\rho} = v^T \dot{w} \quad \dot{v} = S\rho \dot{w} \quad (7)$$

which has to be compared with the typical gradient equations in the $\rho$ and $V_k$ variables given by

$$\dot{\rho} = v^T \dot{w} \quad \dot{v} = \frac{S}{\rho} \dot{w}. \quad (8)$$

The two dynamical systems are thus quite similar, differing by a $\rho^2$ factor in the $\dot{v}$ equations. It is clear that the stationary points of the gradient for the $v$ vectors—that is the values for which $\dot{v} = 0$—are the same in both cases, as for any $t > 0$, $\rho(t) > 0$.

Importantly, the almost equivalence between constrained and unconstrained GD is true only when $p = 2$ in the unit $L_p$ norm constraint[5]. In both cases the stationary point (for fixed but usually very large $\rho$, that is a very long time) are the same and likely to be (local) minima. In the case of deep networks, we expect multiple such minima for any finite $\rho$.

**Convergence to minimum norm and maximum margin.** Consider the flow of $V_k$ in Eq. (6) for fixed $\rho$. If we assume that for $t > T_0$, $f(V; x)$ separates the data, i.e., $\forall n \; y_n f(V)(x_n) > 0$, then $\frac{d}{dt} \rho_k > 0$, i.e., $\rho$ diverges to $\infty$ with $\lim_{t \to \infty} \dot{\rho} = 0$. In the one-layer network case, the dynamics yields $\rho \approx \log t$ asymptotically. For deeper networks, this is different. Banburski et al.[5] shows that the product of weights at each layer diverges faster than logarithmically, but each individual layer diverges slower than in the one-layer case. Banburski et al.[5] also shows that $\dot{\rho}_k^2$ (the rate of growth of $||\rho_k||^2$) is independent of $k$.

The stationary points at which $\dot{V}_k = 0$ for fixed $\rho$ —if they exist—satisfy the necessary condition for a minimum, i.e.,

$$\sum e^{-y_n \rho f_V(x_n)} \left( \frac{\partial f_V(x_n)}{\partial V_k^{i,j}} - V_k^{i,j} f_V(x_n) \right) = 0 \quad (9)$$

are critical points of the loss for fixed $\rho$ (as the domain is compact minima exist). Let us call them $V_k(\rho) = min_{||V_k||=1} \sum_n e^{-y_n \tilde{f}(x_n)}$. Consider now the limit for $\rho \to \infty$

$$\lim_{\rho \to \infty} min_{||V_k||=1} \sum_n e^{-y_n \rho \tilde{f}(x_n)} = \lim_{\rho \to \infty} min_{||V_k||=1} e^{-\rho \tilde{f}(x*)}, \quad (10)$$

where $x^*$ corresponds to the training data that are support vectors and have the smallest margin. This provides a solution $V_k(\infty)$, which corresponds to a maximum margin solution, because $\lim_{\rho \to \infty} V_k(\rho) = \min_n \max_{||V_k||=1} \tilde{f}_\rho(x_n)$ (see also ref. [5]). Those maximum margin solutions are also minimum norm solution in terms of the $W_k$ for a fixed margin $\geq 1$.

In summary, the dynamics of $\dot{V}_k$ for each fixed $\rho$ converges to a critical point which is likely to be a minimum on the boundary of the $\rho$ ball. For $\rho \to \infty$, the stationary points of the full dynamical system Eq. (6) are reached. These points are degenerate equilibria[5].

## Discussion

The main conclusion of this analysis is that unconstrained GD in $W_k$ followed by normalization of $W_k$ is equivalent to imposing a $L_2$ unit norm constraint on $V_k = \frac{W_k(t)}{\|W_k(t)\|}$ during GD. For binary and multiclass classification, only the normalized weights are needed. To provide some intuition, consider that GD is steepest descent wrt the $L_2$ norm and the steepest direction of the gradient depends on the norm. The fact that the direction of the weights converge to stationary points of the gradient under a constraint is the origin of the hidden complexity control, described as implicit bias of GD by Srebro and colleagues[7], who first showed its effect in the special case of linear networks under the exponential loss. In addition to the approach summarized here, another elegant theory[8–10], leading to several of the same results, has been developed around the notion of margin, which is closely related, as in the case of support vector machines, to minimization of an exponential-type loss function under a norm constraint.

In summary, there is an implicit regularization in deep non-linear networks trained on exponential-type loss functions, originating in the GD technique used for optimization. The solutions are in fact the same as that are obtained by vanishing regularized optimization. This is thus similar to—but more robust than—the classical implicit regularization induced by iterative GD on linear networks under the square loss and with appropriate initial conditions. In our case, the maximum margin solutions are independent of initial conditions and the linearity of the network. The specific solutions, however, are not unique, unlike the linear case: they depend on the trajectory of gradient flow, each corresponding to one of multiple minima of the loss, each one being a margin maximizer. In general, each solution will show a different test performance. Characterizing the conditions that lead to the best among the margin maximizers is an important open problem.

The classical analysis of ERM algorithms studies their asymptotic behavior for the number of data $n$ going to infinity. In this limiting classical regime, $n > D$, where $D$ is the fixed number of weights; consistency (informally the expected error of the empirical minimizer converges to the best in the class) and generalization (the empirical error of the minimizer converges to the expected error of the minimizer) are equivalent. The capacity control described in this paper implies that there is asymptotic generalization and consistency in deep networks in the classical regime (see Fig. 1). However, as we mentioned, it has been shown in the case of kernel regression, that there are situations in which there is simultaneously interpolation of the training data and good expected error. This is a modern regime in which $D > n$ but $\gamma = \frac{D}{n}$ is constant. In the linear case, it corresponds to the limit for $\lambda = 0$ of regularization, i.e., the pseudoinverse. It is likely that deep nets may have a similar regime, in which case the implicit regularization described here is an important prerequisite for a full explanation—as it is the case for kernel machines under the square loss. In fact, the maximum margin solutions we characterize here are equivalent to minimum norm solutions (for margin equal to 1, see ref. [5]). Minimum norm of the weight matrices implies minimum uniform stability and thus suggests minimum expected error, see ref. [11]. This argument would explain why deep networks trained with exponential losses predict well and why classification error does not increase with overparametrization (see Fig. 2). It would also explain, in the case of kernel methods and square-loss regression, why the pseudoinverse solution provides good expected error and at the same time perfect interpolation on the training set[12,13] with a data-dependent double-descent behavior.

As we mentioned, capacity control follows from convergence of the normalized weights during GD to a regularized solution with

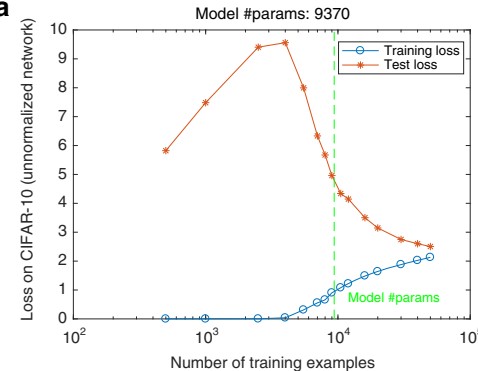

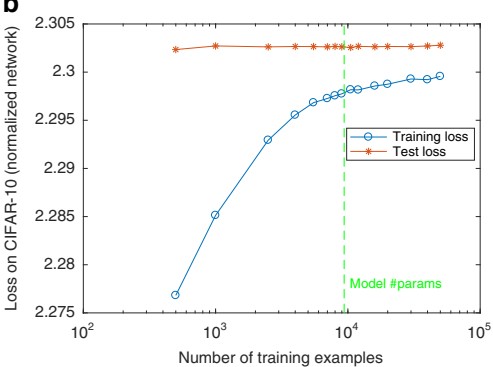

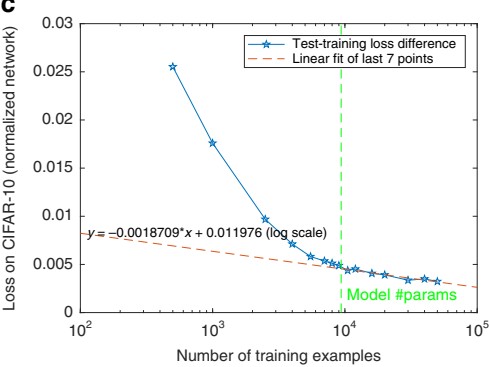

**Fig. 1 Classical generalization and consistency in deep networks.**
**a** Unnormalized cross-entropy loss in CIFAR-10 for randomly labeled data. **b** Cross-entropy loss for the normalized network for randomly labeled data. **c** Generalization cross-entropy loss (difference between training and testing loss) for the normalized network for randomly labeled data as a function of the number of data $N$. The generalization loss converges to zero as a function of $N$ but very slowly.

vanishing $\lambda$. It is very likely that the same result we obtained for GD also holds for SGD in deep networks (the equivalence holds for linear networks[14]). The convergence of SGD usually follows convergence of GD, although rates being different. The Robbins–Siegmund theorem is a tool to establish almost sure convergence under surprisingly mild conditions.

It is not clear whether a similar effective regularization should also hold for deep networks with more than two layers trained with square loss. In fact, we have not been able to find a mechanism that could lead to a similarly robust regularization. Therefore, we conjecture that deep networks trained with square loss (with more than two layers and not reducible to kernels) do not converge to minimum norm solutions, unlike the same networks trained on exponential-type losses.

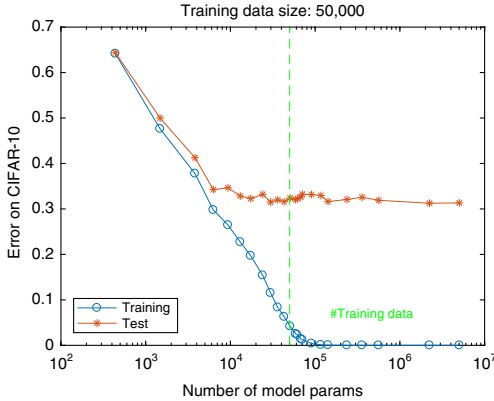

**Fig. 2 No overfitting in deep networks.** Empirical and expected error in CIFAR-10 as a function of number of neurons in a 5-layer convolutional network. The expected classification error does not increase when increasing the number of parameters beyond the size of the training set.

Interestingly, the theoretical observations we have described suggest that the dynamics of $\rho$ may be controlled independently from GD on the $V_k$, possibly leading to faster and better algorithms for training deep networks. A hint of this possibilities is given by an analysis for linear networks (see ref. 5) of the dynamics of weight normalization (Eq. (7)) vs. the dynamics of the unconstrained gradient (Eq. (8)). Under the same simplified assumptions on the training data, the weight normalization dynamics converges much faster—as $\frac{1}{t^{1/2\log t}}$—than the typical dynamics, which converges to the stationary point as $\frac{1}{\log t}$. This prediction was verified with simulations. Together with the observation that $\rho(t)$ associated with Eq. (8) is monotonic in $t$ after separability is reached, it suggests exploring a family of algorithms that consist of an independently driven forcing term $\rho(t)$ coupled with the equation in $V_k$ from Eq. (8).

$$\dot{V}_k = \frac{\rho}{\rho_k^2} \sum_{n=1}^{N} e^{-\rho f_V(x_n)} \left( \frac{\partial f_V(x_n)}{\partial V_k} - V_k f_V(x_n) \right). \quad (11)$$

The open question is of course what is the optimal $\rho(t)$ schedule for converging to the margin maximizer that is best in terms of expected error.

## Data availability
All relevant data are publicly available (Github: https://github.com/liaoq/natcom2020).

## Code availability
All relevant code are publicly available (Github: https://github.com/liaoq/natcom2020).

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

## Acknowledgements
This material is based upon work supported by the Center for Minds, Brains and Machines (CBMM), funded by NSF STC award CCF-1231216, and part by C-BRIC, one of six centers in JUMP, a Semiconductor Research Corporation (SRC) program sponsored by DARPA.

## Author contributions
T.P. developed the basic theory. Q.L. and A.B. run simulations. All three authors wrote the paper.

## Competing interests
The authors declare no competing interests.
