## [Peer Review File · Nature Communications]

Reviewers' comments:

Reviewer #1 (Remarks to the Author):

The paper provides illuminating insights on the self-regularizing properties of DL. However, it leaves the reviewer with a slight unhappiness, as the immediate question that comes: 'now what' remains only discussed a tiny bit.

While I already like the paper as is, I am adding some remarks (some of them random) that come to my mind when reading and integrating the ms into my mind map

1. Opening the embedding to literature, relate or discuss to classical statistical insights such as NIC Murata et al 91 NC, perhaps even relate to Montavon's kernel analysis of DL (Montavon et al 2011 JMLR) or Tali Tishby's new work. It may be illuminating to have an overall view of this.
2. In Raetsch et al 2000 ML, it was shown that Boosting is a gradient decent in the same loss function where the focus is placed through some annealing like process ultimately on the hard to learn patterns. Boosting inevitably overfits unless a regularizer is included into that loss. I understand that keeping the norm of the ensemble weights α_i (when $\sum \alpha_i f_{\theta_i}$ is the ensemble classifier) itself steady did not help avoiding overfitting. The point might however have been to analyse the overall norm of all weights included into this ensemble (if I gather your ms right)?
3. I was a bit frustrated by the end of the paper. It says that the dynamics of ρ may be controlled independently from gradient descent on the V_k . This sounds intriguing but also somewhat unclear what to make of it. I feel that the paper makes a first move to understand things and it would be nice to be a bit more explicit about what can be done practically with that insight so that we could improve things. Demonstrating this would increase the ms' impact hugely and make the reader much happier.
4. Finally, I am missing a mention of the other local minima. I gather that the discussion of this ms holds near a local minimum. Shouldn't there be at least a discussion why this may actually be a good one. SGD folks have already shown this. So aren't there 2 effects, one being the general SGD bringing the model to a reasonable solution and then your argument saying that it is doing that in a self-regularizing manner so that the complexity of the function class remains in bounds... Please clarify.

Reviewer #2 (Remarks to the Author):

This paper discusses some aspects of hidden complexity control in deep neural networks. The paper is well written and the problem addressed is very important in deep learning research. However, there are a number of reasons for which I believe that the paper cannot be recommended for publication (at least not in this form), as follows:

Major comments:

- 1) The proofs have a number of assumptions, which make the overall obtained results quite narrow and not so significant, as the paper claims, e.g. the use of Gradient Descent in Box 1. Similar proofs shall be given for Stochastic Gradient Descent and Batch Gradient Descent which are typically used in Deep Learning.
- 2) The use of exponential loss function. There are many other loss functions, e.g. mean squared error, for other type of tasks, e.g. regression problems, which have to be discussed.
- 3) Large parts of this manuscript (text and mathematical proofs) are similar with Section 3 of "Theoretical Issues in Deep Networks: Approximation, Optimization and Generalization", PNAS, 2019, by partly the same authors, <https://arxiv.org/pdf/1908.09375.pdf> which is not cited. Smaller, but also similar, overlaps are with Ref. 3 which partly has the same authors. This reduces

seriously the novelty of this manuscript. Please address these issues in the revised version.

Minor comments:

4) "there is no obvious control of complexity in the training of deep networks". In fact there are some kind of forms, e.g. the number of parameters gives a form of complexity control... Please rephrase or discuss this.

5) "controlling the complexity". It would be good to define what the authors understand by "complexity" and to add some references to help clarifying this concept.

6) "Standard deep learning training". As far I know there is not a consensus on the standardization of deep learning training. Perhaps, using something like "typical deep learning training" would be more suitable.

7) Equation 1 notations (e.g. W_k) and T , T_0 from the next paragraph are defined in Box 1 and 2. Perhaps it would be good to define them in the main text or in an Appendix specially made for notations.

8) "Box 1". The equations sometimes contain $f(x_n)$ and sometimes $f(W; x_n)$. Is there a difference? If so, please unify/clarify notations. It would be useful to do this unification of notations, in fact, for the whole manuscript.

9) A number of other previous works shall be discussed in this manuscript, as I believe that they are important in this line of research. Below are some non-exhaustive examples:

Michael Lutter, Christian Ritter, Jan Peters, Deep Lagrangian Networks: Using Physics as Model Prior for Deep Learning, ICLR 2019

Behnam Neyshabur, Srinadh Bhojanapalli, David McAllester, Nathan Srebro, Exploring Generalization in Deep Learning, NIPS, 2017

Hongyang Zhang, Junru Shao, Ruslan Salakhutdinov, Deep Neural Networks with Multi-Branch Architectures Are Less Non-Convex, 2018

10) I would recommend to the authors to incorporate all of the above comments in the next version of the manuscript, and to develop further in at least one concrete case the idea from their last paragraph "theoretical observations we have described suggest how the dynamics of ρ may be controlled independently from gradient descent". These may bring enough delta novelty for this manuscript to reach a publishable threshold.

Reviewer #3 (Remarks to the Author):

In this brief paper, Prof. Poggio developed a method of control to speed up the training of deep networks. In particular, he argued that a classical form of norm control, a kind of control of hidden complexity, is responsible for the general working of deep neural networks. The argument was based on the simple observation that the weights computed during minimization of the exponential loss do not matter by themselves, and in fact they diverge during gradient descent. For example, in the case of classification with deep nonlinear networks, control of complexity can be done with respect to the direction of the weights when the normalized weights are defined as the variables of interest. In particular, a weight vector is written as the product of its norm and a unit vector specifying its direction. The weight evolution equation is accordingly replaced by two dynamical equations, one for the norm and another for the unit vector specifying the direction of the original

weight. Implicit complexity control is achieved by solving the equation for the unit direction of the weight. The speculation is that this mechanism underlies "hidden" complexity control in deep networks and likely represents the basic reason for their generalization properties.

With all due respect, I cannot recommend this paper for publication in Nature Communications, for the following reasons.

1. (Major) The heuristic argument leading to the speculation that the evolution of the unit direction of the weight represents "hidden" complexity control in deep networks and likely represents the basic reason for their generalization properties is in fact quite speculative. The idea and argument could have been more convincing had there been some numerical examples in the paper. Especially, concrete examples demonstrating that the new gradient descent strategy based on the unit direction vector outperforms the classical gradient descent method are needed.
2. (Major) Why applying gradient descent on the unit direction vector of the weight would lead to fast convergence? How much faster in comparison with the standard approach? Is there any physical intuition?
2. (Minor) The concept of hidden complexity, its precise definition and characterization are not introduced in a comprehensive manner, making it difficult to understand the paper.
3. (Minor) The style of the presentation seems to deviate from the norm of a typical paper in Nature Communications. For experts in deep learning, the brief paper may be OK. However, for a broad audience, the paper can be very difficult and unfriendly.
4. (Minor) There is a typo in the second formula in Eq. (3).

NatureCommunication: manuscript NCOMMS-19-31086-T

Answers to reviewers comments

Reviewers' comments:

Reviewer #1 (Remarks to the Author):

The paper provides illuminating insights on the self-regularizing properties of DL. However, it leaves the reviewer with a slight unhappiness, as the immediate question that comes: 'now what' remains only discussed a tiny bit.

While I already like the paper as is, I am adding some remarks (some of them random) that come to my mind when reading and integrating the ms into my mind map

1. Opening the embedding to literature, relate or discuss to classical statistical insights such as NIC Murata et al 91 NC, perhaps even relate to Montavon's kernel analysis of DL (Montavon et al 2011 JMLR) or Tali Tishby's new work. It may be illuminating to have an overall view of this.

2. In Raetsch et al 2000 ML, it was shown that Boosting is a gradient decent in the same loss function where the focus is placed through some annealing like process ultimately on the hard to learn patterns. Boosting inevitably overfits unless a regularizer is included into that loss. I understand that keeping the norm of the ensemble weights α_i (when $\sum \alpha_i f_{\theta_i}$ is the ensemble classifier) itself steady did not help avoiding overfitting. The point might however have been to analyse the overall norm of all weights included into this ensemble (if I gather your ms right)?

The papers you suggest are actually quite interesting for developing aspects of my work in the future, especially wrt the connection with boosting and margin but are not directly relevant to the work described. The note I submitted however was not intended to be a review of theoretical results on deep nets: a recent paper cites from 2012 to 2019 at least 313 references. I focused and cited only the ones that are most directly relevant to complexity control in the minimization of an exponential-type loss by RELU deep networks and are also at the forefront of theoretical understanding of deep nets. Needless to say, there is no paper yet with a satisfactory full theory of deep networks.

3. I was a bit frustrated by the end of the paper. It says that the dynamics of rho may be controlled independently from gradient descent on the V_k . This sounds intriguing but also somewhat unclear what to make of it. I feel that the paper makes a first move to understand things and it would be nice to be a bit more explicit about what can be done practically with that insight so that we could improve things. Demonstrating this would increase the ms' impact hugely and make the reader much happier.

This is a good point. I described an example of two different dynamics in the linear case in a drastically revised "Conclusion" section.

4. Finally, I am missing a mention of the other local minima. I gather that the discussion of this ms holds near a local minimum. Shouldn't there be at least a discussion why this may actually be

a good one. SGD folks have already shown this. So aren't there 2 effects, one being the general SGD bringing the model to a reasonable solution and then your argument saying that it is doing that in a self-regularizing manner so that the complexity of the function class remains in bounds... Please clarify.

Again, yes, you are absolutely right, there should be a discussion and I have added it. I do not think the SGD folks have proven theoretically that minimizers are good minima in terms of the expected error (though there are several conjectures such as flat minima). In an online memo I have argued years ago, that SGD selects degenerate minima which tend to be global but the arguments are not yet formal proofs.

Reviewer #2 (Remarks to the Author):

This paper discusses some aspects of hidden complexity control in deep neural networks. The paper is well written and the problem addressed is very important in deep learning research. However, there are a number of reasons for which I believe that the paper cannot be recommended for publication (at least not in this form), as follows:

Major comments:

1) The proofs have a number of assumptions, which make the overall obtained results quite narrow and not so significant, as the paper claims, e.g. the use of Gradient Descent in Box 1. Similar proofs shall be given for Stochastic Gradient Descent and Batch Gradient Descent which are typically used in Deep Learning.

I agree that it would be good to have similar proofs for SGD. I added a comment on it. Weight normalization — which is close to Batch Normalization — is included in my note (it corresponds to constrained gradient descent).

2) The use of exponential loss function. There are many other loss functions, e.g. mean squared error, for other type of tasks, e.g. regression problems, which have to be discussed.

Yes, I agree. I added a comment why exponential loss functions are especially interesting. The proofs cannot be easily extended to square loss regression.

3) Large parts of this manuscript (text and mathematical proofs) are similar with Section 3 of "Theoretical Issues in Deep Networks: Approximation, Optimization and Generalization", PNAS, 2019, by partly the same authors, <https://arxiv.org/pdf/1908.09375.pdf> which is not cited. Smaller, but also similar, overlaps are with Ref. 3 which partly has the same authors. This reduces seriously the novelty of this manuscript. Please address these issues in the revised version.

There are indeed overlaps with online arxiv papers as you correctly pointed out. I think — and I hope to be correct — that it is Nature policy that online papers are not considered an obstacle to submitting to Nature journals. The online version “Theoretical Issues in Deep Networks: Approximation, Optimization and Generalization” is old: it reflects a talk I gave at the NAS which was based on Ref. 3. A much different version may appear as a review in a special issue of PNAS not before the second quarter of 2020. If it will appear, there will be references there to this paper, which, if accepted, will appear much earlier. Ref 3 will remain only online for the foreseeable future.

Minor comments:

4) “*there is no obvious control of complexity in the training of deep networks*”. *In fact there are some kind of forms, e.g. the number of parameters gives a form of complexity control... Please rephrase or discuss this.*

5) “*controlling the complexity*”. *It would be good to define what the authors understand by “complexity” and to add some references to help clarifying this concept.*

I agree about 4 and 5. I made changes to the ms accordingly.

6) “*Standard deep learning training*”. *As far I know there is not a consensus on the standardization of deep learning training. Perhaps, using something like “typical deep learning training” would be more suitable.*

Good suggestion that I have implemented!

7) *Equation 1 notations (e.g. W_k) and T, T_0 from the next paragraph are defined in Box 1 and 2. Perhaps it would be good to define them in the main text or in an Appendix specially made for notations.*

Done!

8) “*Box 1*”. *The equations sometimes contain $f(x_n)$ and sometimes $f(W;x_n)$. Is there a difference? If so, please unify/clarify notations. It would be useful to do this unification of notations, in fact, for the whole manuscript.*

Done!

9) *A number of other previous works shall be discussed in this manuscript, as I believe that they are important in this line of research. Below are some non-exhaustive examples:*

Michael Lutter, Christian Ritter, Jan Peters, Deep Lagrangian Networks: Using Physics as Model Prior for Deep Learning, ICLR 2019

Behnam Neyshabur, Srinadh Bhojanapalli, David McAllester, Nathan Srebro, Exploring Generalization in Deep Learning, NIPS, 2017

Hongyang Zhang, Junru Shao, Ruslan Salakhutdinov, Deep Neural Networks with Multi-Branch Architectures Are Less Non-Convex, 2018

These are interesting references but there are many other ones at the same level of relevance for this note. The note I submitted is not a review of theoretical results on deep nets: there must be several hundreds theoretical papers on deep nets in the last 6 years. I focused only on the ones that are most directly relevant to complexity control during minimization of an exponential-type loss by RELU deep networks. Needless to say, there is no paper yet with a satisfactory full theory of deep networks!

10) I would recommend to the authors to incorporate all of the above comments in the next version of the manuscript, and to develop further in at least one concrete case the idea from their last paragraph "theoretical observations we have described suggest how the dynamics of ρ may be controlled independently from gradient descent". These may bring enough delta novelty for this manuscript to reach a publishable threshold.

I tried to add an example to develop the idea about independent control of ρ .

Reviewer #3 (Remarks to the Author):

In this brief paper, Prof. Poggio developed a method of control to speed up the training of deep networks. In particular, he argued that a classical form of norm control, a kind of control of hidden complexity, is responsible for the general working of deep neural networks. The argument was based on the simple observation that the weights computed during minimization of the exponential loss do not matter by themselves, and in fact they diverge during gradient descent. For example, in the case of classification with deep nonlinear networks, control of complexity can be done with respect to the direction of the weights when the normalized weights are defined as the variables of interest. In particular, a weight vector is written as the product of its norm and a unit vector specifying its direction. The weight evolution equation is accordingly replaced by two dynamical equations, one for the norm and another for the unit vector specifying the direction of the original weight.

Implicit complexity control is achieved by solving the equation for the unit direction of the weight. The speculation is that this mechanism underlies "hidden" complexity control in deep networks and likely represents the basic reason for their generalization properties.

With all due respect, I cannot recommend this paper for publication in Nature Communications, for the following reasons.

- 1. (Major) The heuristic argument leading to the speculation that the evolution of the unit direction of the weight represents "hidden" complexity control in deep networks and likely represents the basic reason for their generalization properties is in fact quite speculative. The idea and argument could have been more convincing had there been some numerical examples in the paper. Especially, concrete examples demonstrating that the new gradient descent strategy based on the unit direction vector outperforms the classical gradient descent method are needed.*

The note describes an implicit complexity control in the *existing* gradient descent algorithms currently in use. It does not propose a new strategy. I tried to clarify this point in the new version, especially by defining more carefully the specific goals of the paper in the introduction.

2. (Major) *Why applying gradient descent on the unit direction vector of the weight would lead to fast convergence? How much faster in comparison with the standard approach? Is there any physical intuition?*

See above.

2. (Minor) *The concept of hidden complexity, its precise definition and characterization are not introduced in a comprehensive manner, making it difficult to understand the paper.*

I agree. I specifically tried to explain better this point in a completely new introduction.

3. (Minor) *The style of the presentation seems to deviate from the norm of a typical paper in Nature Communications. For experts in deep learning, the brief paper may be OK. However, for a broad audience, the paper can be very difficult and unfriendly.*

I changed the format hoping that the paper is clearer...

4. (Minor) *There is a typo in the second formula in Eq. (3).*
thanks!

REVIEWERS' COMMENTS:

Reviewer #1 (Remarks to the Author):

I feel the revision has improved clarity and I am now less unhappy after reading than in the original submission. I suggest to accept as is.

Reviewer #2 (Remarks to the Author):

I thank the author for partly considering my previous comments. Indeed, the paper is improved in comparison with the last version. Its level of clarity grew up. Still, as acknowledged also by the author, the results are still narrow. The comments introduced about gradient descent and other type of loss functions do not necessarily replace the proofs for those cases. Also, as mentioned by Reviewer 3, some numerical results would make the paper more convincing.

On a side note, submitting papers with (perhaps partly) similar content in terms of novel results to be reviewed in two places at same time (i.e. Nature Communications and PNAS) is not the best practice. At this moment, it is difficult to analyze the overlap between the two papers as also the PNAS paper seems to be under review. Indeed, arXiv papers shall be fine, as far I know.

Minor comment: $V_{\{}}$, and $\rho_{\{k}}$ in Section 2.2 first paragraph are not introduced, making them a bit unclear. It is true that they are explained/used in the next paragraph.

To conclude, I believe that this paper does not have the required level of novelty to be accepted for publication in Nature Communications. Thus, unfortunately, I cannot recommend its acceptance.

Reviewer #3 (Remarks to the Author):

The author has addressed all referee comments. The revised paper is more readable than the previous version. I recommend acceptance.

NatureCommunication: manuscript NCOMMS-19-31086-T
Answers to reviewers comments (second iteration, Dec 2019)

Reviewers' comments:

Reviewer #2 (Remarks to the Author):

I thank the author for partly considering my previous comments. Indeed, the paper is improved in comparison with the last version. Its level of clarity grew up. Still, as acknowledged also by the author, the results are still narrow. The comments introduced about gradient descent and other type of loss functions do not necessarily replace the proofs for those cases. Also, as mentioned by Reviewer 3, some numerical results would make the paper more convincing.

On a side note, submitting papers with (perhaps partly) similar content in terms of novel results to be reviewed in two places at same time (i.e. Nature Communications and PNAS) is not the best practice. At this moment, it is difficult to analyze the overlap between the two papers as also the PNAS paper seems to be under review. Indeed, arXiv papers shall be fine, as far I know.

Minor comment: $V_{\{j\}}$, and $\rho_{\{k\}}$ in Section 2.2 first paragraph are not introduced, making them a bit unclear. It is true that they are explained/used in the next paragraph.

To conclude, I believe that this paper does not have the required level of novelty to be accepted for publication in Nature Communications. Thus, unfortunately, I cannot recommend its acceptance.

As we mention in the text our results are likely to extend to SGD (this seems however to require a separate technical paper). The extension to square loss is much less obvious and may not even be true. In fact, the reviewer comments prompted us to state a conjecture about lack of complexity control in very deep networks trained under the square loss. As we mention in the text, the exponential-type loss functions are the most relevant because they are the ones used most commonly in practice. In particular, the cases in which deep nets are clearly better than other techniques — such as in ImageNet, speech recognition etc. — all involve classification problems and thus training with cross-entropy.

The PNAS article, if it will be published, will be purely a review of work done in our group. This review will include and cite original work described in this Nature Communication submission (if accepted). Let me emphasize again that this is a particular situation because of a special and very late PNAS issue collecting material from a 2019 NAS workshop.